# Connecting Air Pollution Exposure to Socioeconomic Status: A Cross-Sectional Study on Environmental Injustice among Pregnant Women in Scania, Sweden

**DOI:** 10.3390/ijerph16245116

**Published:** 2019-12-14

**Authors:** Erin Flanagan, Emilie Stroh, Anna Oudin, Ebba Malmqvist

**Affiliations:** 1Division of Occupational and Environmental Medicine, Department for Laboratory Medicine, Lund University, 222 42 Lund, Skåne, Sweden; erin.flanagan@med.lu.se (E.F.); emilie.stroh@med.lu.se (E.S.); anna.oudin@med.lu.se (A.O.); 2Environmental Medicine, Department for Public Health and Clinical Medicine, Umeå University, 901 87 Umeå, Västerbotten, Sweden

**Keywords:** ambient air pollution, nitrogen oxide, particulate matter, NO_X_, PM_2.5_, socioeconomic status, environmental injustice, pregnant women

## Abstract

Environmental injustice, characterized by lower socioeconomic status (SES) persons being subjected to higher air pollution concentrations, was explored among pregnant women in Scania, Sweden. Understanding if the general reduction of air pollution recorded is enjoyed by all SES groups could illuminate existing inequalities and inform policy development. “Maternal Air Pollution in Southern Sweden”, an epidemiological database, contains data for 48,777 pregnancies in Scanian hospital catchment areas and includes births from 1999–2009. SES predictors considered included education level, household disposable income, and birth country. A Gaussian dispersion model was used to model women’s average NO_X_ and PM_2.5_ exposure at home residence over the pregnancy period. Total concentrations were dichotomized into emission levels below/above respective Swedish Environmental Protection Agency (EPA) Clean Air objectives. The data were analyzed using binary logistic regression. A sensitivity analysis facilitated the investigation of associations’ variation over time. Lower-SES women born outside Sweden were disproportionately exposed to higher pollutant concentrations. Odds of exposure to NO_X_ above Swedish EPA objectives reduced over time, especially for low-SES persons. Environmental injustice exists in Scania, but it lessened with declining overall air pollution levels, implying that continued air quality improvement could help protect vulnerable populations and further reduce environmental inequalities.

## 1. Introduction

Environmental injustice or inequity describes the disproportionate distribution of environmental contaminants within areas populated by socially, economically, or otherwise disadvantaged groups [1]. Due to the inherent nature of ambient air pollution, deleterious pollutants are concentrated along major traffic sites or upwind from emitter points, such as factories or energy-production complexes [2]. These more heavily polluted areas tend to hold lower housing market prices [3,4] that persons of low socioeconomic status (SES) can afford. Furthermore, such low land prices are more likely to attract additional pollutive land-use decisions, including roadway and industry development, than upmarket areas [3]. A contrasting pattern has also emerged, whereby reconstruction and revitalization attract high-income earners to settle in more heavily polluted inner-city areas [3]. Conversely, those commuting into cities in their personal vehicles predominantly live in suburban areas [3,5], meaning they are less affected by the traffic pollution they themselves create [5]. Identifying vulnerable subpopulations, including those of low SES, who are more exposed to high air pollution concentrations is vital to mitigating environmental injustice. The majority of studies on socioeconomic indicators and air pollution uncovered an association between low SES and exposure to high levels of air pollution [5,6,7,8,9,10,11,12,13,14,15], although not always consistently [16,17]. Moreover, a 2015 global review that evaluated and summarized 37 environmental justice articles [1] indicated that studies from North America, New Zealand, Asia, and Africa typically found higher air pollution levels in low-SES areas, while the findings of European studies were mixed [1]. As previously mentioned, this is likely because some European cities feature historic centers that attract high-income earners despite the elevated air pollution levels. Because associations can differ both within cities and across countries, results often depend upon the particular city, as well as on the geographical scale chosen [18,19]. Some agreement exists, however, that areas across Europe characterized by lower SES tend to have greater concentrations of air pollution [20,21].

Countless epidemiological studies have been conducted on the variety of effects that air pollution exposure, particularly nitrogen dioxide (NO_2_), has on human health. For instance, previous studies demonstrated associations between exposure to air pollution and respiratory disease [22,23,24], cardiovascular disease [25,26,27], and neurodevelopmental and cognitive disorders [28,29], among others. Moreover, the World Health Organization’s comprehensive 2013 Review of evidence on health aspects of air pollution (REVIHAAP) provided confirmation of a causal link between exposure to particulate matter (PM) with an aerodynamic diameter less than or equal to 2.5 µm (PM_2.5_) and adverse health outcomes, including morbidity and mortality, cardiovascular disease, childhood respiratory disease, cognitive development, and, most pertinent to this study, birth outcomes [30]. With this, particulate matter’s short- and long-term effects on human health are reinforced, and their continued investigation is warranted.

Likely due to their higher exposure, more deprived living conditions, and copious stressors, persons of low SES are often more likely to develop these various air pollution-related diseases than their higher-SES counterparts [31,32]. Pregnant women have also been identified as a vulnerable population [33,34,35,36]. The oxidative stress and systemic inflammation invoked by air pollution was shown to contribute to pregnancy complications in both experimental and observational studies [37,38,39]. Associations between exposure to air pollution and gestational diabetes and preeclampsia were also demonstrated in epidemiological studies [40,41,42,43]. Furthermore, an extensive review of 11 studies in the United States (USA), comprising over one million pregnant women, demonstrated evidence of significant, positive associations between PM and hypertensive disorders of pregnancy (HDP) [36]. These maternal conditions can endanger the pregnancy and lead to premature birth or low-birth-weight babies [44]. Air pollution exposure has also been linked directly to preterm birth [45], fetal growth restriction [33], small for gestational age [46], and many other fetal growth indicators [8,47]. These metrics are considered predictors for the child’s well-being; in fact, worse growth in utero, termed “fetal programming”, is a risk factor for the later development of disease [48]. 

Recognizing air pollution as a major public health issue, the European Union (EU) set air pollution thresholds for its member-nations [49], and the World Health Organization (WHO) acknowledged the need for stricter targets for PM in their air quality guidelines [50]. These are presently under revision, and the WHO is expected to publish their new guidelines in 2020 [50]. Despite this, concentrations below current EU and WHO thresholds were shown to have harmful health effects, especially for children, those already sick, pregnant women, and their unborn babies, as well as other susceptible groups [20,33,43,46,47,51,52]. For this reason, the Swedish Environmental Protection Agency (EPA) set its own limits in the Clean Air objective, which aims for even lowerNO_2_ targets (see Table 1) [53]. 

Certainly, the relationship between SES and air pollution is often more complex than implied and can vary among SES measurements chosen, pollutants considered, and study areas explored [1]. This present study was performed in Sweden, a nation considered to be rather egalitarian, with an advanced welfare system and relatively low income inequality (Gini index of 29.20 in 2015) [54]. Despite this, two articles within the same study area demonstrated an association between low SES and exposure to high levels of air pollution [19,55]. For instance, this relationship’s variability among five Scanian cities and the region as a whole was described by Stroh et al. [19], and Chaix et al. reported an association between SES and exposure to air pollution among children in Malmö, both at home and at school [55].

This prior research [19,55] explored relationships from 2001; thus, an updated investigation is needed with more current data now available. Furthermore, neither of these previous studies focused on pregnant women, a more susceptible population, nor did they include PM exposure in their analyses. As the harmful effects of air pollution’s many components on maternal health [40,41,42,43] and fetal development [33,46,47,51], even below EU and WHO air quality thresholds, are well documented, it is vital to learn of pregnant women’s exposure in Scania, a relatively low-exposure setting. Air pollution levels in this region underwent a general decline in recent years [56], yet whether associations between air pollution exposure and SES have varied over time remains to be explored. This was, in fact, identified as a priority research gap [1] that could help explain if all SES groups similarly experienced and enjoyed the overall reduction of air pollution. A stronger understanding of whether the unequal distribution of air pollution is becoming wider or narrower could subsequently inform policy development needed to realize environmental justice.

## 2. Materials and Methods

### 2.1. Study Population

MAPSS (Maternal Air Pollution in Southern Sweden), an epidemiological database, consists of a cohort of 48,777 pregnancies provided by Perinatal Revision Syd (PRS), a local birth register [51]. The majority (76–86%) of women in this cohort were between 26 and 40 years old (see Appendix A). The MAPSS database includes the major hospital catchment area of Scania, which are hereafter referred to as the total study population or the entire catchment area. Virtually all (98–99%) births occurring in this area between 1999 and 2009 are included. Furthermore, this study separately investigates MAPSS women living in the Malmö and Lund municipalities. An in-depth description of the original birth cohort was provided by Malmqvist et al. [51]. 

### 2.2. Study Setting

The large hospital catchment area this study explores is situated in Scania (Skåne), the southernmost county of Sweden, as well as two cities (Malmö and Lund) located within the region itself (see Figure 1). Scania covers approximately 11,350 km^2^ and is one of the most heavily populated regions, where about 11% of the Swedish population (around 1.1 million people) resides [19,57]. Although this region’s air pollution levels are lower than other, more densely populated areas of the world, lying well below the EU air quality guidelines and mostly under targets proposed by WHO [51], adverse health outcomes have been identified even at low exposure concentrations. This is especially true for vulnerable populations, such as pregnant women and their babies [33,43,46,51,58]. Furthermore, Scania is of particular interest as it has relatively high levels of air pollution compared to the rest of Sweden [19,57]. A large portion of road users from the European continent, including both passenger cars and freight trucks, travel on Scania’s numerous motorways as they continue throughout Sweden and also into Norway [19,57]. Moreover, several harbors along the coast expose the region to emissions from continuous cargo shipping and ferry transport [19,57]. Scania’s proximity to Copenhagen, Denmark and the rest of Zealand, combined with prevailing westerly winds, also contributes to the high concentrations of air pollutants recorded [19,57]. The two cities chosen for this study are located in close proximity to one another in the southwestern part of Scania. Malmö, the largest city in Scania and third largest in Sweden overall, is located on the west coast. It has approximately 322,000 inhabitants, spans 77 km^2^, and is considered one of Sweden’s most segregated cities [19,59,60]. In contrast, Lund is an inland university town with a population of around 116,000 covering 23 km^2^. 

### 2.3. Data Collection

#### 2.3.1. Socioeconomic Status

Variables used to determine socioeconomic status (education level, household disposable income, and birth country) were derived from Statistics Sweden and linked to MAPSS through each participant’s personal identification number. Categories encompassing birth country included “Sweden”, “other Nordic countries”, “other European Union member countries (EU-28)”, “other European countries”, “North America”, “South America”, “Africa”, and “Asia”. Certain groups were excluded from the investigation: 40 from “Oceania”, as conclusions could not be drawn from this small size, and eight pregnancies without maternal country of birth information. 

Women’s education level at the year they gave birth was gathered and separated into “low”, “medium”, and “high” educational attainment. It should be noted that this included only the highest completed education. Low education corresponds those who attended primary schooling for nine years or less; medium is defined as a finished secondary education; and high constitutes women with completed post-secondary degrees.

Household disposable income (HDI), including social benefits, was divided into four categories: “30,000–200,000”, “200,000–300,000”, “300,000–400,000” and “>400,000” Swedish kronor (SEK) per year. This separation closely represents the quartile distribution of income among the total study population. To deduce the lower limit of 30,000 SEK annually, the EU’s definition of a person at risk of poverty was employed, i.e., those living in a household with a disposable income below 60% of the country’s median value [61]. The lowest median household disposable income reported by Statistics Sweden that aligns with this project’s study population was for 20–29-year-old women living alone: 149,700 SEK per year [62], making those with an annual HDI of 59,880 SEK at risk of poverty. Thus, setting the lowest value at 30,000 SEK per year was deemed reasonable to include a wide range of low-income households.

#### 2.3.2. Air Pollution Exposure

Nitrogen oxide (NO_X_) is the collective term for the presence of nitrogen monoxide (NO) and nitrogen dioxide (NO_2_). Agencies and governments often choose to regulate NO_2_, as seen in Table 1, because it is the most prevalent component of NO_X_ found in the atmosphere. This is because any NO initially present transforms into NO_2_ over time [63]. With NO_2_ nearly synonymous with NO_X_, this study uses the combined definition NO_X_ for its assessment of air pollution exposure. For reference, a rough conversion estimate of average annual NO_X_ values to NO_2_ was calculated and included in the Appendix A using the following formula [63]:NO_2_ = NO_X_^(0.75 + (18/(NO_x_ + 60)))^.(1)

(A) Emissions Database (EDB)

The EDB utilized contains over 25,000 local emission sources, including road traffic, railways, shipping, major industrial sites, non-road vehicles, small-scale energy and heat producers, aviation, and the contribution of air pollution from Denmark. Specific road traffic emissions were derived from the Swedish Road Administration’s data concerning vehicle type, fuel source, traffic intensity, and speed limits [57]. As a few diesel-powered freight trains remain in an otherwise electrified system, railroad emissions in the study area were estimated from these trains’ fuel consumption [63]. Emissions attributable to shipping were appraised by Gustafsson [63] and Project Shipair [64]. Data on major industrial sites’ air pollution contribution were gathered from Emission Register (EMIR), a national database [57]. Air pollution emissions from non-road vehicles, such as construction machinery, were reported by the Swedish Environmental Research Institute (IVL) [65]. Furthermore, small-scale energy and heat production emissions were obtained from the National Rescue Agency, where the frequency of stove use could be predicted using chimney sweep records [63]. Annual environmental reports from Scandinavian airports provided aviation emission data [57]. Because the largest Danish island, Zealand, neighbors Scania and westerly winds prevail, Zealand’s local emissions, including that of the industrial capital city, Copenhagen, were incorporated [57]. A more detailed account of these emission source categories was previously discussed by Malmqvist et al. [57]. 

(B) Exposure Modeling

A modified Gaussian, flat, two-dimensional dispersion model (AERMOD), a United States Environmental Protection Agency model, was constructed using the software program ENVIMAN [63] and used to model air pollution exposure for the 1999–2009 study period. Modeling was based on local meteorology and the comprehensive EDB detailed above. Furthermore, a high temporal resolution (hourly) was applied. Nitrogen oxide (NO_X_) was modeled for each year utilizing two databases with differing spatial resolutions; births occurring in 1999–2005 were paired with 500 m grid cells, while a spatial resolution of 100 × 100 m was used for birth years 2006–2009. Due to the high cost and time demands of modeling emissions for such a large region, concentrations of PM_2.5_ were modeled for two years, 2000 and 2011, with 100-m grids. Using a linear model, data for the years in between were interpolated, and an atmospheric ventilation index was applied to account for seasonal dispersion variation. The resulting PM_2.5_ data for 2000–2009 were then utilized in this study. Further details of this process were presented by Malmqvist et al. [57]. Both the data and the dispersion model were thoroughly documented and systematically validated against actual measurements of air pollutants with respect to concentration levels, as well as the appropriateness of spatial and temporal resolutions [57,63].

To account for non-local PM transported into the region, measured levels of PM_2.5_ at an urban meteorological background station were used in the dispersion model [57]. The total PM_2.5_ levels used in this project were, thus, the sum of locally emitted PM_2.5_ from the EDB and far-reaching background PM_2.5_ transported into the study area. Total NO_X_ levels were calculated by adding 2.5 µg/m^3^, the average levels calculated at a rural background site, to local NO_X_ emissions from the EDB.

The geographic coordinates of each woman’s residential address were obtained through her unique personal identification number and used to assess individual air pollution exposure.

(C) Exposure Variables

The modeled hourly concentrations were amassed into monthly mean levels. These were first compiled into pregnancy trimester averages, which were then aggregated further to encompass mean exposure over the entire pregnancy period. The resulting total PM_2.5_ variable was finally dichotomized into emission levels below and above its corresponding Swedish EPA Clean Air objective, i.e., 10 µg/m^3^ [53]. Because 2.5 µg/m^3^ of background NO_X_ was added to the local levels, its cut-off was set at 17.5 µg/m^3^ to equal the Swedish EPA objective of 20 µg/m^3^ [53]. SES groups subjected to pollutant concentrations above Clean Air objectives may hereafter be referred to as high exposure or highly exposed.

### 2.4. Statistical Analyses

The data were analyzed with SPSS Statistical Software version 25 (IBM^®^, Armonk, NY, USA) using binary logistic regression. Associations between exposure to total NO_X_ and total PM_2.5_ above the Clean Air objectives and socioeconomic predictors were explored and described with odds ratios (ORs) with 95% confidence intervals (CIs). This was performed for each study setting (the entire catchment area, Malmö, and Lund), enabling comparison across a wider area and between two distinct municipalities. For this analysis, high education, the second highest income quartile, and Sweden were selected as reference categories for education level, household disposable income, and birth country, respectively. Moreover, a sensitivity analysis was performed separating the birth years into two groups (1999–2005 and 2006–2009), facilitating the investigation of environmental injustice’s variation over time. The two databases for modeled NO_X_ having these year groupings necessitated the same separation here.

To facilitate a more direct comparison to Stroh et al.’s country of birth variable and corresponding results, birth country was dichotomized into “Sweden” and “other countries” in an additional analysis for NO_X_ in Lund. A further sensitivity analysis on parity, or the number of times a woman has given birth, was performed to assess uncertainty involving HDI misclassification.

### 2.5. Ethical Approval

The Lund University Ethical Committee approved this study (Registration number: 696/2014). 

## 3. Results

### 3.1. Main Findings

During this study period, women in the total MAPSS catchment area were, on average, exposed to 17.8 µg/m^3^ of NO_X_ and 11.1 µg/m^3^ of PM_2.5_ (see Table 2). Examples of exposure levels from dispersion modeling are illustrated in Figure A1 and Figure A2 for NO_X_ and PM_2.5_, respectively (see Appendix B). Mean levels of both pollutants were highest in Malmö, especially NO_X_, and lowest in Lund. Descriptive information regarding the three study populations and socioeconomic status indicators can be found in Table 3. Concerning highest achieved education, 12% of women in the entire catchment area only completed nine years or less of primary school; this rose to 19% in Malmö and dropped to as little as 5% in Lund. Again, a higher proportion (36%) of women belonged to the lowest household disposable income quartile in Malmö than in Lund (15%). Swedish-born women comprised the majority (66%) of the total study population, constituted only half of Malmö’s study population, and represented the overwhelming majority in Lund (80%).

In the total MAPSS study population (Table 4), women with low educational attainment experienced higher odds of being exposed to levels of both pollutants exceeding Clean Air objectives than those possessing a post-secondary degree: 2.71 (95% CI: 2.54–2.88) and 1.34 (95% CI: 1.20–1.50) for NO_X_ and PM_2.5_, respectively. Furthermore, women living with annual household disposable incomes (HDIs) in the lowest quartile had greater odds of being highly exposed, nearly four times for NO_X_ and around double for PM_2.5_, compared to those in the second highest HDI quartile. Odds ratios (ORs) were seen to decrease with increasing HDI, with the exception of the highest HDI quartile for PM_2.5_. In the total study population, the odds of being exposed to NO_X_ and PM_2.5_ levels beyond Clean Air objectives were greater among all women not born in Sweden but to varying degrees. Women born in Africa, Asia, European countries outside the EU, and South America, for instance, had the greatest odds of being highly exposed compared to Swedish-born women. 

Results pertaining to education level and household disposable income from the municipality of Malmö mirrored the main findings, albeit to a lesser degree and without a stepwise exposure effect gradient for HDI and NO_X_ (Table A1, Appendix B). Concerning birth country in Malmö, there was a tendency, although not statistically significant, for other Nordic-born women to have lower odds of being highly exposed than those born in Sweden (Table A1, Appendix B). The trends seen in the entire catchment area also continued in the same direction for women in Lund with regard to HDI, but education level ORs diverged slightly with higher-educated women having greater odds of being highly exposed to PM_2.5_ than their less educated counterparts (Table A2, Appendix B). Furthermore, associations between a woman’s country of birth and exposure to air pollutants above the Clean Air objectives in Lund varied, and most failed to uphold statistical significance. The majority of findings for PM_2.5_ exposure in Malmö and Lund were statistically insignificant or presented wide CIs.

### 3.2. Sensitivity Analyses: Change over Time, Birth Country and Parity

In the entire catchment area, mean NO_X_ levels dropped from 18.6 µg/m^3^ from 1999–2005 to 16.2 µg/m^3^ from 2006–2009, but PM_2.5_ concentrations remained virtually unchanged (Table A3, Appendix C). These trends continued in Malmö and Lund (Table A3, Appendix C). Regarding the study population’s demographic variation over the study period, the percentage of women with high education and high income increased from 1999 to 2009 in all study settings, yet changes in women’s birth regions varied by setting (Table A4, Appendix C).

Results in Table A5 (Appendix C) indicate that odds of being exposed to concentrations of NO_X_ above Clean Air objectives decreased over the study period for those of the total study population with low educational attainment, those within the lowest annual HDI quartile, and for virtually all foreign-born women, especially women born in “other European countries”, “Africa”, and “Asia”. Conversely, odds indicating high exposure to PM_2.5_ seem to have increased over the study period (Table A5, Appendix C). Women living in Malmö and Lund experienced the same tendency as those in the entire catchment area with few exceptions, albeit with statistical insignificance (Table A6 and Table A7, respectively, Appendix C). In Lund, opposite patterns for education level, more varied results for birth country, and a decline in odds of PM_2.5_ exposure above the Clean Air objective were documented (Table A7, Appendix C).

Concerning the dichotomization of birth country, results remained consistent with Table A2 (Appendix B) despite this adjustment (Appendix A). In the sensitivity analysis on parity, comparing women with only one child to those with >1 child gave no indication that this misclassification drove the study’s main results (Appendix A). 

## 4. Discussion

### 4.1. Main Findings

Generally, the pregnant women included in this study were exposed to levels of air pollution below the EU’s current average annual air quality guidelines [49], yet WHO’s more stringent limit values on yearly means of PM_2.5_ were exceeded [50]. Further from being met was the Swedish EPA’s Clean Air objective, with stricter annual average limits for NO_2_ [53]. The Swedish EPA claims that these objectives were modified to consider and protect vulnerable populations and to ensure that air pollution concentrations remain below even low-risk levels for disease development and environmental protection [53], employing the precautionary principle.

As per the Clean Air objective, PM_2.5_ target values were consistently surpassed in each study setting and at every time period, to the largest extent in Malmö and the least in Lund. The NO_2_ Clean Air objective, however, was only exceeded in Malmö (Appendix A). Malmö is a much larger, more populated municipality, with its own harbor for cargo and leisure seagoing vessels. Furthermore, being a coastal town, it lies closer to shipping routes, as well as to industrial Zealand. Malmö is also a highly trafficked city with vehicle transit moving through it via the Öresunds bridge. The compilation of these factors likely contributes to the greater annual means of NO_X_ and PM_2.5_ present in Malmö as compared to Lund and to the aggregated, diluted levels seen throughout the entire catchment area. 

This study’s evidence of unequally distributed environmental hazards among different socioeconomic groups is indicative that environmental injustice exists in Scania, Sweden. Indeed, women born in regions comprising lower- and middle-income countries (LMIC), such as Africa, Asia, other European countries, and South America, tended to have substantially greater odds than Swedish-born women, while women from other high-income countries (HIC), such as other Nordic nations, European Union member countries, and North America were not as highly exposed. Furthermore, the stepwise reduction of odds as education and income increased, with few exceptions, mimicked the social gradient of health. Because exposure to high levels of air pollution is associated with adverse health outcomes, one can postulate that increasing SES provides progressively greater protection against exposure and, thus, against poorer health outcomes. With this, the burden of air pollution-related illness would be disproportionately experienced by pregnant women with little education, low income, and born in LMIC in Lund, Malmö, and the entire MAPSS catchment area.

However, Stroh et al. cautioned against generalizing results gleaned from a sizeable region, such as this study’s hospital catchment area, as relationships specific to larger cities within it may skew the results [19]. As Malmö comprises a large proportion of the entire study population, it may be driving the regional results. Furthermore, immigrants to Sweden typically settle in cities where pollution levels are higher, as opposed to the countryside [19]. With this, a rural–urban gradient could contribute to the strong associations seen between birth country and all pollutants in the entire catchment area. Comparing the two municipalities, odds of being highly exposed were typically greater for women in Malmö than in Lund. It is plausible that this is because a larger proportion of the study population in Malmö had low education, low household disposable income, and a birthplace outside Sweden. Malmö is even considered one of Sweden’s most segregated cities [19]. The distinct demographic compositions of these two cities, as well as their differentially exposed locations mentioned previously, presumably contributed to the varying results seen. 

These findings are partially consistent with Stroh et al.’s study on correlations between socioeconomic characteristics and exposure to air pollution in 2001 among the general population in Scania, Sweden. For instance, the authors found that being an immigrant (i.e., not Swedish-born) living in Scania and Malmö was associated with higher concentrations of NO_2_ [19]. Similarly, this study suggests that all women not born in Sweden experienced greater odds of being highly exposed to NO_X_ compared to Swedish-born women in the total study population and in Malmö. Stroh et al. discovered the opposite trend when examining the city of Lund: being born in Sweden was associated with elevated levels of NO_2_ [19]. The results presented here, however, are varied and far from statistically significant. Interestingly, the only statistically significant odds ratio found for birth country in Lund contradicted Stroh et al.’s findings and mirrored our other study settings: women from “other European” countries had greater odds of high NO_X_ exposure compared to Swedish-born women. The overall lack of association here could indicate that this variable may not be a suitable indicator of one’s exposure to air pollution in Lund. When birth country was dichotomized to mimic Stroh et al.’s study, the findings aligned with our own results. This may be because associations for different birth regions varying in direction negated each other when merged.

Education level was also considered by Stroh et al. When dichotomizing low vs. high educational achievement, the investigators found that less educated persons were exposed to greater concentrations of NO_2_ in Scania, as well as in Malmö [19]. Parallel to this, the odds of experiencing NO_X_ levels above the Clean Air objective in the entire catchment area and Malmö were higher for women with low and even medium education levels. Again, the inverse was found by Stroh et al. for Lund [19]. Odds derived in this study are not completely in agreement with Stroh et al., as lower-educated women in Lund still had greater odds of being exposed to NO_X_ concentrations than more highly educated women. These findings were under the cut-off for statistical significance, indicating that an association between education level and NO_X_ exposure may no longer be present in Lund. As with the majority of birth country categories, it appears that women with varying degrees of education were more proportionately spread throughout Lund’s more and less NO_X_-polluted areas. 

Chaix et al. previously investigated socioeconomic status in terms of mean annual income, examining exposure to NO_2_ among children throughout the Malmö municipality in 2001 [55]. Their results illustrated that NO_2_ levels consistently increased with decreasing socioeconomic status, both at home (for both building-specific and neighborhood income levels) and at school [55]. Although MAPSS is connected to each woman’s individual household disposable income, this study’s results for the Malmö population are in line with Chaix et al.’s. Specifically, a gradient of exposure can be seen with one’s odds of being highly exposed to NO_X_ decreasing with increasing HDI. Unfortunately, Chaix et al.’s study focused solely on the city of Malmö; thus, comparisons can neither be drawn for the entire catchment area overall, nor for Lund specifically. Interestingly, however, HDI was the only SES variable with viable results concerning women residing in the Lund municipality. While Stroh et al. did not directly investigate income, they attributed the unique association in Lund of higher NO_2_ exposure among Swedish-born, high-educated persons to a trend common in urban areas: persons with higher economic status living in costly, sought-after apartments in city centers where air pollution levels are typically higher [19]. Again, our results illustrated the opposite. This could be attributed to air pollution levels increasing where low-income women tend to live, a possible reduction of air pollution in Lund’s city center, the movement of more wealthy individuals to less polluted suburbs, or a number of other factors and their intertwined effects. 

Considering a wider, international perspective, a recent multi-city study from the EURO-HEALTHY (Shaping European Policies to Promote Health Equity) project explored air pollution variability (NO_2_) according to various socioeconomic indicators in nine European metropolitan areas, including Stockholm, Sweden [18]. It is important to note that comparisons of environmental injustice between different settings is difficult because of each one’s distinct composition; however, some trends can be seen. Stronger associations of NO_2_ exposure were found among areas with a larger proportion of foreign-born inhabitants, specifically outside the European Union (EU-28), with higher unemployment rates, and with a greater number of total crimes per 100,000 inhabitants [18]. Authors observed these associations to be linear and highly statistically significant [18]. Our study similarly reports a stepwise gradient of increasing odds of being highly exposed with increasing SES deprivation, lending support to the authors’ indication that persons living in more disadvantaged urban areas of Europe experience poorer air quality [18]. 

As no previous literature exists regarding particulate matter exposure and socioeconomic status within this study setting, studies from other European countries were considered. To begin, higher concentrations of various PM fractions were detected in neighborhoods composed of >20% non-white persons in England and non-Western immigrants the Netherlands [66].This trend continued among low-SES persons, specifically including low education, in an Italian study [2]as well as among the most deprived neighborhoods in England [16,66] and France [9]. These European studies correspond to our presented results, especially throughout the entire MAPSS study population and Malmö, with few deviations. The SES indicators used in this study did not yield many significant results in relation to PM exposure in Lund, as was the case for NO_X_. However, an exception can be seen with low- and medium-educated women in Lund experiencing lower odds of high PM_2.5_ exposure compared to the highest-educated women. Although a different pollutant, this adheres to Stroh et al.’s results in Lund [19]. Such findings illustrate the complexity of air pollution research, and they are an indication that associations can be pollutant- or source-specific. 

Another study investigating environmental injustice among 16 city cohorts from eight Western European countries through the European Study of Cohorts for Air Pollution Effects (ESCAPE) project reported conflicting results based upon the SES indicator chosen [67]. Persons living in neighborhoods with greater unemployment, for instance, were subjected to higher concentrations of NO_2_, whereas those personally belonging to a lower socioeconomic group appeared to be less exposed [67]. Contrary to the latter, we found lower individual household disposable income to be associated with increased odds of high exposure, indicating that the neighborhood and individual differences described by these authors may not be applicable to the Scanian setting. In Malmö, this could potentially be explained by residual effects of the Million program, a substantial low-income housing initiative implemented from 1965–1974 [68]. These buildings were erected in higher-exposure zones, including industrial areas [55] and between Malmö’s system of heavily trafficked roads and highways [19]. Attempting to resolve the housing crisis, this city planning decision disproportionately exposed low-SES residents to a greater environmental burden. This oversight, having a lasting effect on environmental justice, should be considered by policymakers as an example of unintended consequences to be avoided in the future.

### 4.2. Sensitivity Analysis: Change over Time

Following the general decline of NO_X_ over the study period, women with low education, within the lowest HDI quartile, and from LMIC saw their odds drop more substantially than their highly educated, more affluent, and HIC counterparts throughout the entire catchment area and in Malmö. This suggests that decreases in NO_X_ pollution levels benefitted low-SES persons the most, and it is indicative of a decrease in disparities on an absolute basis.

Clark et al. observed similar findings in their study on the changes of air pollution exposure by “race/ethnicity” and SES from 2000 to 2010 in the United States [69]. They explained that the reduction of NO_2_ disparities between non-white and white participants could be greatly attributed to overall drops in ambient NO_2_ levels, specifically from new vehicle emission-control technology and decreases in industry emissions [69]. As Stroh et al. described, the majority of immigrants typically settle into cities, where NO_2_ levels are generally higher [19], and neighborhoods in Malmö consisting of higher proportions of non-Swedish-born persons are often situated on the outskirts, where the city’s five main highways converge [19]. Chaix et al. also mentioned the proximity of pollutive facilities to underprivileged areas of Malmö [55]. This information would suggest that the considerable reduction of odds for NO_X_ exposure above the Clean Air objective would be dependent upon traffic-related and industry-sourced reductions overall, as well as near low-SES women’s homes in Malmö and, to some extent, the total study population.

Conversely, PM_2.5_ concentrations remained stagnant from 1999–2009. This likely is because high portions of PM are transported in; thus, local or regional reduction efforts may have been ineffective. Also potentially adding to its constant levels was authorities’ encouragement of residents to utilize small-scale heating devices, a large emitter of PM_2.5_, during the study period. Without a reduction in pollution levels, odds of being highly exposed to PM_2.5_ rose in the entire catchment area and Malmö, especially among low-SES persons. With this, a widening of environmental injustice was discovered. These findings suggest that general air pollution reduction is critical for the mitigation of exposure inequalities. Results from this sensitivity analysis for both pollutants were more varied in Lund and should be regarded with caution as sample sizes reduced greatly, yielding statistical insignificance and wide confidence intervals. Because of this, well-grounded conclusions cannot be drawn from this setting.

Overall, this sensitivity analysis indicates that environmental justice relating to NO_X_ exposure improved over time (1999–2009) in Scania. Despite this, low-SES women’s odds of exposure to pollutant concentrations exceeding Clean Air objectives remained considerably greater at virtually every time period studied relative to those of higher SES. These persisting exposure disparities cannot be ignored. It is imperative that efforts to reduce general levels of air pollution throughout Scania, Sweden, especially urban areas, are continued. As seen with PM_2.5_, the consequences of unsuccessful reduction efforts can disproportionately affect persons of lower socioeconomic status, leading to greater environmental injustice. Moreover, focusing policy and intervention efforts in areas characterized by lower SES would further mitigate environmental inequalities. Such measures would also help ensure that fewer women would experience air pollutants at concentrations that are harmful to their pregnancies and the development of their babies, both in utero and later in life.

With the Swedish Environmental Research Institute (IVL) approximating roughly 7,600 deaths to be caused by NO_2_, PM_10_, and PM_2.5_ exposure each year in Sweden [70], the overall potential health gain following air pollution reduction is undeniable. It was estimated that 2–4% of total annual deaths could be prevented and several morbidities reduced in the city of Malmö alone with the theoretical removal of traffic-related air pollution [71]. Furthermore, IVL estimated the costs of air pollution exposure to be approximately 56 billion SEK in 2015 [70]. Such evidence corroborates that the continued decrease of air pollution emissions is vital for the protection of public health, as well as the safeguarding of limited healthcare and financial resources.

### 4.3. Methodological Considerations

The three socioeconomic variables were each incorporated into a separate logistic regression, as detangling their effects on one another is complex. For instance, one may inquire whether country of birth lies in the causal pathway to income increase, or if it could influence the level of education one achieves. Furthermore, a person’s education is often linked to their income; thus, adjusting for both simultaneously could result in overfitting of the model. How these SES indices correlate with one another was examined. Relatively low correlations demonstrate that they account for different phenomena (Appendix A). 

A vital strength to this study is its large sample population from the MAPSS birth cohort, recording 98–99% of all births taking place in major hospital catchment areas throughout Scania, Sweden. Further, the employment of a thoroughly validated, high-resolution (both temporally and spatially) dispersion model allowed for the vetted exposure estimates of NO_X_ and PM_2.5_. The emission database it is based upon utilizes over 25,000 points of exposure data and is continually updated, amassing a level of detail that improves the reliability of results. Gathering geographic points of women’s exposure at home derived from a dependable source was also essential for the precision and validity of exposure evaluation. The accurate and impartial register data for all socioeconomic status variables from Statistics Sweden also eliminated recall and response bias, enhancing validity. Results from this study are deemed generalizable to the wider populations of Lund, Malmö, and the entire catchment area, as pregnant women are similarly exposed when not pregnant and reside alongside other populations. The odds of exposure to air pollution levels above Clean Air objectives presented here could likely be generalized to other persons of low socioeconomic status within the study settings. Finally, this study is the first in Scania, Sweden to investigate the relationship between socioeconomic status and PM_2.5_ exposure, include pregnant women as the study population, and explore the change of these associations over time. 

Despite these assets, limitations are also present. To begin, discrepancies relating to the different databases and spatial resolutions for NO_X_ from 1999–2005 (500 m × 500 m) and 2006–2009 (100 m × 100 m) were present. It is difficult to ascertain whether the declining average concentrations of NO_X_ seen in Table A3 (Appendix C) are due to actual reductions, the differing databases used, or a combination of the two. However, measurements derived from a coarser-scale grid, such as that utilized from 1999–2005, more likely underpredict the true exposure [72]. This lends support to the reported NO_X_ emissions reflecting actual reductions. Importantly, these are still considered high spatial resolutions able to capture exposure measures, and their suitability was validated [51,57].

Furthermore, a systematic error is present within the household disposable income variable. A “household” is defined by Statistics Sweden as a married couple who may have any number of children. However, if the cohabitating parents of a singular child are unmarried, only the mother’s income is considered. This holds until the woman is pregnant with a second child, at which point her partner’s income is also included. Hence, it is possible that unmarried women with only one child were misclassified into a lower annual HDI quartile if they live with an earning significant other. This could result in an underestimation of the true odds low-income women face. The sensitivity analysis conducted reinforced that this misclassification did not drive the study’s main results (Appendix A). On the contrary, if primiparous women were excluded, a stronger correlation between HDI and air pollution would likely be seen. 

Misclassification error is a concern innate to air pollution studies. Here, ambient air pollution exposure was modeled at the point of residence, which is standard practice. Indoor, occupational, as well as transport- and behavior-related exposures were not evaluated, as this study did not seek to assess total exposure. Instead, it aimed to investigate potential differential exposure at home. Nevertheless, the utilization of reliable residential data and functional dispersion models is essential for decreasing bias [19]. Both conditions were considered to be satisfied. Moreover, the smallest study location unit was the municipal level, meaning the two cities investigated (Malmö and Lund) were not further separated into distinct neighborhoods that might experience varying levels of air pollution. Consequently, some measurement error may have occurred and decreased precision [73]. This can, fortunately, be counteracted by a large enough sample size [73], which this study, with 98–99% of all births in Malmö (20,226 pregnancies) and Lund (8554 pregnancies), likely had. 

### 4.4. Future Research

Further research involving more recent years could provide an additional understanding of how the relationship between air pollution and socioeconomic status has developed in Scania, Sweden, as well as if environmental justice has continued to improve. This could be particularly intriguing in Lund, as it experienced significant growth. As cities are not homogeneous units, it would have been informative to divide the study settings by their distinct neighborhoods and evaluate environmental injustice within and between them.

## 5. Conclusions

Environmental injustice exists in Scania, Sweden, as low-SES women have disproportionate odds of NO_X_ and PM_2.5_ exposure above the Swedish EPA’s Clean Air objective compared to their higher-SES counterparts, with few exceptions. Additionally, PM_2.5_ levels outlined by the Clean Air objective were exceeded throughout Scania, and NO_2_ levels were surpassed in Malmö during the study period. The extensive evidence of air pollution’s effect on the general population’s health, as well as that of pregnant women and their babies, encourages the actualization of these more stringent thresholds in order to protect more susceptible populations and uphold the precautionary principle. This would further contribute to the mitigation of the pervasive morbidity and mortality attributable to air pollution. 

General reductions in NO_X_ over the study period appear to have benefitted lower-SES women to a greater extent than women of higher socioeconomic status, demonstrating that continued improvements in overall air quality may further reduce socioeconomic inequalities in air pollution exposure. Eliminating these disparities altogether, however, would likely require interventions targeted specifically toward more vulnerable populations and the neighborhoods in which they reside, such as disadvantaged urban areas. Evidence of adverse “fetal programing” from prenatal air pollution exposure that extends throughout the life course justifies the classification of pregnant women, especially those of low SES disproportionately exposed, as a priority group with respect to policy development initiatives. 

## Figures and Tables

**Figure 1 ijerph-16-05116-f001:**
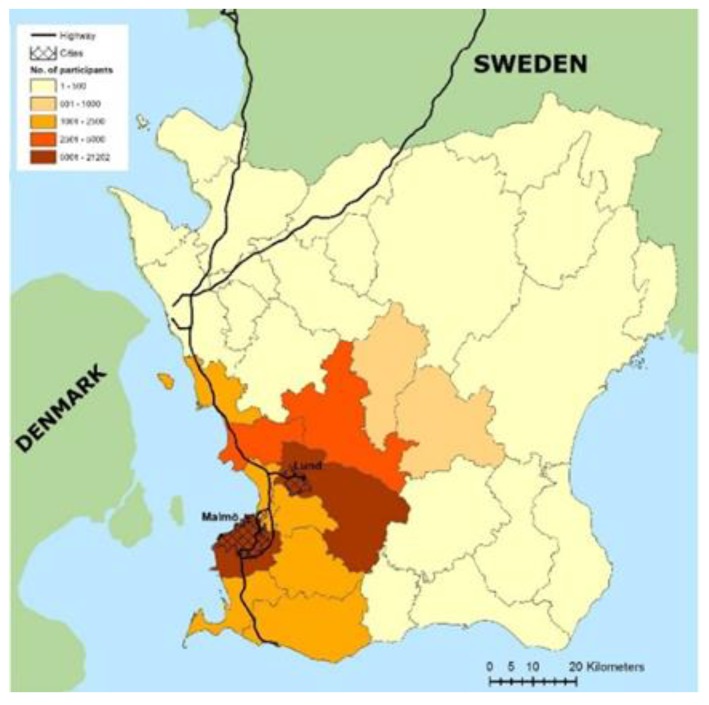
Maternal Air Pollution in Southern Sweden (MAPSS) study population distribution (1999–2009). Map by Emilie Stroh.

**Table 1 ijerph-16-05116-t001:** Average annual air pollution emission limits by organization. EU—European Union; WHO—World Health Organization; EPA—Environmental Protection Agency.

	NO_2_	PM_2.5_
EU	40 µg/m^3^	25 µg/m^3^
WHO	40 µg/m^3^	10 µg/m^3^
Swedish EPA	20 µg/m^3^	10 µg/m^3^

**Table 2 ijerph-16-05116-t002:** Summary of pollutants: average air pollution concentrations (µg/m^3^) over entire pregnancy by each study setting for the study period (1999–2009).

	N (%)	Mean	SD	Range	Median
Total catchment area					
NO_X_	43,670 (90%)	17.8	10.4	3.5–50.2	16.9
PM_2.5_	30,576 (63%)	11.1	1.2	6.9–17.3	11.0
Malmö					
NO_X_	18,666 (92%)	23.6	8.9	4.4–45.3	23.2
PM_2.5_	13,478 (67%)	11.6	1.1	7.4–17.3	11.5
Lund					
NO_X_	8049 (94%)	14.5	7.3	4.1–42.7	14.7
PM_2.5_	5285 (62%)	10.9	1.1	6.9–15.4	10.7

NO_X_ = nitrogen oxide. PM_2.5_ = particulate matter with diameter ≤2.5 µm. SD = Standard deviation.

**Table 3 ijerph-16-05116-t003:** Descriptive information of socioeconomic indicator frequency among the Maternal Air Pollution in Southern Sweden (MAPSS) total study population, Malmö, and Lund.

	Total Study Population (*n* = 48,777)	Malmö (*n* = 20,226)	Lund (*n* = 8554)
	% (N)	% (N)	% (N)
Education level			
Low	12.3% (5991)	18.8% (3808)	5.0% (429)
Medium	40.1% (19,543)	39.8% (8050)	28.9% (2474)
High	39.8% (19,415)	33.7% (6823)	64.8% (5543)
Total	92% (44,949)	92.4% (18,681)	98.7% (8446)
Missing	7.8% (3828)	7.6% (1545)	1.3% (108)
Household disposable income *			
30,000–200,000	22.6% (11,034)	36.3% (7347)	15.2% (1296)
200,000–300,000	24.2% (11,784)	28.2% (5705)	24.0% (2057)
300,000–400,000	25.1% (12,239)	18.6% (3772)	28.2% (2416)
Greater than 400,000	22.1% (10,777)	13.3% (2684)	31.7% (2708)
Total	94.0% (45,834)	96.5% (19,508)	99.1% (8477)
Missing	6.0% (2943)	3.5% (718)	0.9% (77)
Birth country			
Sweden	66.3% (32,335)	50.1% (10,133)	79.6% (6813)
Other Nordic	2.2% (1068)	2.7% (550)	1.9% (166)
Other EU-28	4.3% (2077)	5.7% (1154)	4.6% (393)
Other European	6.6% (3208)	11.2% (2268)	2.8% (237)
North America	0.4% (191)	0.4% (81)	0.6% (50)
South America	1.1% (531)	1.6% (328)	1.1% (94)
Africa	2.3% (1138)	4.2% (847)	1.8% (155)
Asia	12.9% (6292)	23.9% (4832)	7.3% (627)
Total	96.0% (46,840)	99.8% (20,193)	99.8% (8535)
Missing	4.0% (1937)	0.2% (33)	0.2% (19)

* Measured in Swedish kronor (SEK), annually.

**Table 4 ijerph-16-05116-t004:** Associations between exposure to levels of NO and PM_2.5_ (µg/m^3^) above Clean Air objectives ^a^ and socioeconomic status indices for the 48,777 pregnant women of MAPSS living in the entire catchment area.

	OR (95% CI)	
	NO_X_	PM_2.5_
Education level		
Low	2.71 (2.54–2.88) **	1.34 (1.20–1.50) **
Medium	1.29 (1.24–1.35) **	0.85 (0.80–0.91) **
High	REF	REF
Household disposable income ^b^		
30,000–200,000	3.70 (3.50–3.92) **	2.12 (1.92–2.34) **
200,000–300,000	2.21 (2.09–2.34) **	1.36 (1.24–1.48) **
300,000–400,000	REF	REF
>400,000	0.60 (0.56–0.64) **	1.17 (1.08–1.27) **
Birth country		
Sweden	REF	REF
Other Nordic	1.52 (1.33–1.74) **	1.58 (1.26–1.98) **
Other EU-28	2.07 (1.89–2.28) **	1.88 (1.58–2.25) **
Other European	3.68 (3.40–3.98) **	2.97 (2.51–3.52) **
North America	1.54 (1.13–2.10) *	1.86 (1.05–3.31) *
South America	2.77 (2.31–3.32) **	2.21 (1.53–3.19) **
Africa	4.49 (3.94–5.12) **	4.43 (3.13–6.26) **
Asia	4.14 (3.90–4.39) **	3.48 (3.04–3.98) **

^a^ Swedish EPA’s Clean Air objectives, NO_2_: 20 µg/m^3^ and PM_2.5_: 10 µg/m^3^ [53]. ^b^ Measured in Swedish kronor (SEK), annually. OR = odds ratio. CI = confidence interval. NO_X_ = nitrogen oxide. PM_2.5_ = particulate matter with diameter ≤2.5 µm. REF = reference category. * *p* ≤ 0.05. ** *p* ≤ 0.01.

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
