# Peer review of "Connecting Air Pollution Exposure to Socioeconomic Status: A Cross-Sectional Study on Environmental Injustice among Pregnant Women in Scania, Sweden"

_ijerph, 2019, doi:10.3390/ijerph16245116_

Round 1

Reviewer 1 Report

1. The literature review part is a little weak, the author has not given a relatively comprehensive review on the existing literature, and therefore it is hard to highlight the contribution of this research comparing with existing research.

Any data should have its source, so many data has no data source, and this is not acceptable for a Journal paper.

I think the authors are missing some important literature in different sections of the article.

Sun H*., Gulzara T., Haris M., Mohsin M. Evaluating the environmental effects of economic openness: evidence from SAARC countries, Environmental Science and Pollution Research. 2019,6.

Author Response

Thank you for these suggestions. You will find our responses detailed below:

The literature review part is a little weak, the author has not given a relatively comprehensive review on the existing literature, and therefore it is hard to highlight the contribution of this research comparing with existing research.

Thank you for this input, we have now added more literature to the Introduction:

Page 1-2:

Line 44. “The majority of studies on socioeconomic indicators and air pollution has uncovered an association between low SES and exposure to high levels of air pollution [5-15], though not always consistently [16, 17].”

New literature:

15. Verbeek, T., Unequal residential exposure to air pollution and noise: A geospatial environmental justice analysis for Ghent, Belgium. SSM Popul Health, 2019. 7: p. 100340.

Page 2:

Line 51. “Thus, results depend upon the particular city as well as on the geographical scale chosen because associations can differ both within cities and across countries [18, 19].”

Line 65. “Likely due to their higher exposure, more deprived living conditions and copious stressors, persons of low SES often more likely to develop these various air pollution-related diseases than their higher SES counterparts [31, 32]. Pregnant women have also been identified as a vulnerable population [33-36].”

Line 71. “Further, an extensive review of 11 studies in the U.S., comprising over 1 million pregnant women, demonstrated evidence of significant, positive associations between PM and hypertensive disorders of pregnancy (HDP) [36].”

Line 75. “Air pollution exposure has also been linked directly to preterm birth [45], fetal growth restriction [33], small for gestational age [46], as well as many other fetal growth indicators [8, 47].”

New literature:

18. Samoli, E., et al., Spatial variability in air pollution exposure in relation to socioeconomic indicators in nine European metropolitan areas: A study on environmental inequality. Environ Pollut, 2019. 249: p. 345-353.

31. Jans, J., P. Johansson, and J.P. Nilsson, Economic status, air quality, and child health: Evidence from inversion episodes. J Health Econ, 2018. 61: p. 220-232.

32. Mueller, N., et al., Socioeconomic inequalities in urban and transport planning related exposures and mortality: A health impact assessment study for Bradford, UK. Environ Int, 2018. 121(Pt 1): p. 931-941.

36. Koman, P.D., et al., Examining Joint Effects of Air Pollution Exposure and Social Determinants of Health in Defining "At-Risk" Populations Under the Clean Air Act: Susceptibility of Pregnant Women to Hypertensive Disorders of Pregnancy. World Med Health Policy, 2018. 10(1): p. 7-54.

45. Deguen, S., et al., Using a Clustering Approach to Investigate Socio-Environmental Inequality in Preterm Birth-A Study Conducted at Fine Spatial Scale in Paris (France). Int J Environ Res Public Health, 2018. 15(9).

As well as in the Discussion:

Page 10:

Line 399: “Considering a wider, international perspective, a recent multi-city study from the EURO-HEALTHY (Shaping European Policies to promote Health Equity) project explored air pollution variability (NO2) according to various socioeconomic indicators in nine European metropolitan areas, including Stockholm, Sweden [18]. Stronger associations of NO2 exposure were found among areas with a larger proportion of foreign-born inhabitants, specifically outside the European Union (EU28); with higher unemployment rates; and with a greater number of total crimes per 100,000 inhabitants [18]. Authors observed these associations to be linear and highly statistically significant [18]. Our study similarly reports a stepwise gradient of increasing odds of being highly exposed with increasing SES deprivation, lending support to Samoli et al.’s indication that persons living in more disadvantaged urban areas of Europe experience poorer air quality [18].”

New literature:

18. Samoli, E., et al., Spatial variability in air pollution exposure in relation to socioeconomic indicators in nine European metropolitan areas: A study on environmental inequality. Environ Pollut, 2019. 249: p. 345-353.

Page 11:

Line 425: “Another study investigating environmental injustice among 16 city cohorts from 8 Western European countries through the European Study of Cohorts for Air Pollution Effects (ESCAPE) project reports conflicting results based upon the SES indicator chosen [67]. Persons living in neighborhoods with greater unemployment, for instance, were subjected to higher concentrations of NO2; whereas those personally belonging to a lower socioeconomic group were less exposed [67]. Contrary to the latter, we found lower individual household disposable income to be associated with increased odds of high exposure. This could indicate that neighborhood and individual differences described by Temam et al. may not be applicable to the Scanian setting. In Malmö, this could potentially be explained by the Million program, a substantial low-income housing initiative implemented from 1965- 1974 [68]. These buildings were erected in higher exposure zones, including industrial areas [55] and between Malmö’s system of heavily trafficked roads and highways [19]. Attempting to resolve the housing crisis, this city planning decision disproportionately exposed low SES residents to a greater environmental burden. This oversight, having a lasting effect on environmental justice, should be considered by policy makers as an example of unintended consequences to be avoided in the future.”

New literature:

67. Temam, S., et al., Socioeconomic position and outdoor nitrogen dioxide (NO2) exposure in Western Europe: A multi-city analysis. Environ Int, 2017. 101: p. 117-124.

68. Boverket. Under miljonprogrammet byggdes en miljon bostäder. [Internet] 20 May 2014 [cited 2019 5 December]; Available from: https://www.boverket.se/sv/samhallsplanering/stadsutveckling/miljonprogrammet/.

Any data should have its source, so many data has no data source, and this is not acceptable for a Journal paper.

We have read through the manuscript and added data sources where they appeared to be missing. Please feel free to share any further line numbers you feel are missing references, so we can amend this problem.

We also apologize for the error messages ("Error! Reference not found.") throughout the manuscript. An error previously occurred when referencing our own table and figure numbers, but you will now find all these data sources properly named in the Word-file attached.

Thank you again for this input.

I think the authors are missing some important literature in different sections of the article.

Sun H*., Gulzara T., Haris M., Mohsin M. Evaluating the environmental effects of economic openness: evidence from SAARC countries, Environmental Science and Pollution Research. 2019,6. 

As previously mentioned, an updated literature search was conducted and several relevant articles were included in the updated manuscript throughout the Introduction and Discussion sections.

Also, thank you for sharing this article; it seems very interesting. However, we struggle to find the relevance of it in relation to our particular manuscript, but we will keep it for future input.

Reviewer 2 Report

The paper is well written and easy to follow.It makes a sound contribution to the field of environmental justice and environmental policy management. 

I just want to ask the authors to check the manuscript for spelling error. One I foud ocurred on line 247, the vowell e is missing in the word entire. I also wanto to ask the authors to add miising references in the text. i can mention error on lines 109, 114, 241, 242, 243, etc.

Author Response

Thank you very much for taking the time to review this paper and provide input.

I just want to ask the authors to check the manuscript for spelling error. One I found occurred on line 247, the vowel e is missing in the word entire.

We have checked the error for the word "entire" and other such issues, and this seems to be a line-separation error due to a formatting problem that occurred when the Word document was translated to a PDF (i.e. the "e" is stands alone on the very end of the line above the rest of the word "ntire" in the PDF version). We will upload the Word-file again and contact the editor with help to fix this problem. 

The manuscript has also been reviewed for general spelling errors.

I also wan to to ask the authors to add missing references in the text. i can mention error on lines 109, 114, 241, 242, 243, etc.

I apologize for this error message ("Error! Reference source not found.") that occurred in the manuscript when naming data references, such as our table and figure numbers. This has now been fixed. 

Reviewer 3 Report

This manuscript assumes that particulate matter (PM) has important mortality effects in general and particularly for PM2.5.  If such effects do not exist there is less reason to speculate that there is a problem with environmental injustice in Southern Sweden. 

Two recent articles make the case for the absence of mortality effects from PM in general and from PM2.5 in particular.  Your manuscript does not discuss this possibility and its implications for your conclusion that environmental injustice exists in Southern Sweden, but I find the arguments for it simple but compelling. They rest in large part on simple observations rather than statistics.  I think these arguments need to be discussed since it could result in very different conclusions for the paper and for how economically desirable reductions of the use of coal are for climate purposes. This is of importance in many countries, including the US, China, and India.

This different viewpoint is currently at the center of the debate on PM and PM2.5 in the US, and this may actually be of more importance than a discussion of environmental injustice in Southern Sweden.  The articles come from a US blog, but that does not mean that they are incorrect or not worth considering.  If there are no mortality effects from PM and PM2.5, there cannot be much environmental injustice from differing levels  of PM2.5 in different parts of Sweden.

The manuscript appears to rely on the environmental standards of nearby countries to argue the relevance of the pollutant concentrations discussed.  But if these standards as well as those in the US are not based on sound science, this is of little relevance.  

I do not think this issue needs to result in extensive rewriting of the manuscript, just mention and discussion in the introduction of the paper and perhaps elsewhere as a major uncertainty in the conclusions of the paper. 

Author Response

Thank you for taking your valuable time for this revision:

This manuscript assumes that particulate matter (PM) has important mortality effects in general and particularly for PM2.5. If such effects do not exist there is less reason to speculate that there is a problem with environmental injustice in Southern Sweden.

Two recent articles make the case for the absence of mortality effects from PM in general and from PM2.5 in particular.  Your manuscript does not discuss this possibility and its implications for your conclusion that environmental injustice exists in Southern Sweden, but I find the arguments for it simple but compelling. They rest in large part on simple observations rather than statistics.  I think these arguments need to be discussed since it could result in very different conclusions for the paper and for how economically desirable reductions of the use of coal are for climate purposes. This is of importance in many countries, including the US, China, and India.

This different viewpoint is currently at the center of the debate on PM and PM2.5 in the US, and this may actually be of more importance than a discussion of environmental injustice in Southern Sweden.  The articles come from a US blog, but that does not mean that they are incorrect or not worth considering.  If there are no mortality effects from PM and PM2.5, there cannot be much environmental injustice from differing levels of PM2.5 in different parts of Sweden.

The manuscript appears to rely on the environmental standards of nearby countries to argue the relevance of the pollutant concentrations discussed.  But if these standards as well as those in the US are not based on sound science, this is of little relevance. 

I do not think this issue needs to result in extensive rewriting of the manuscript, just mention and discussion in the introduction of the paper and perhaps elsewhere as a major uncertainty in the conclusions of the paper.

The scientific consensus behind the morbidity, mortality and numerous health effects of PM2.5 are strong, and we argue against the reviewer’s opinion that we should downplay these effects. PM2.5’s effects were reviewed by WHO in 2005 and again in 2013, in which the 2013 WHO Europe review of the evidence, often referred to as REVIHAAP, states: 

“Since the 2005 global update of the WHO air quality guidelines (WHO Regional Office for Europe, 2006) were issued, many new studies from Europe and elsewhere on both short- and long-term exposure to PM with an aerodynamic diameter smaller than 2.5 µm (PM2.5) have been published. These studies provide considerable support for the scientific conclusions in the 2005 global update of the WHO air quality guidelines and suggest additional health outcomes to be associated with PM2.5. Among the major findings to date are the following:

1. additional support for the effects of short-term exposure to PM2.5 on both mortality and morbidity, based on several multicity epidemiological studies;

2. additional support for the effects of long-term exposures to PM2.5 on mortality and morbidity, based on several studies of long-term exposure conducted on large cohorts in Europe and North America;

3. an authoritative review of the evidence for cardiovascular effects, conducted by cardiologists, epidemiologists, toxicologists and other public health experts, concluded that long-term exposure to PM2.5 is a cause of both cardiovascular mortality and morbidity;

4. significantly more insight has been gained into physiological effects and plausible biological mechanisms that link short- and long-term PM2.5 exposure with mortality and morbidity, as observed in epidemiological, clinical and toxicological studies;

5. additional studies linking long-term exposure to PM2.5 to several new health outcomes, including atherosclerosis, adverse birth outcomes and childhood respiratory disease; and

6. emerging evidence that also suggests possible links between long-term PM2.5 exposure and neurodevelopment and cognitive function, as well as other chronic disease conditions, such as diabetes.

The scientific conclusions of the 2005 global update of the WHO air quality guidelines about the evidence for a causal link between PM2.5 and adverse health outcomes in human beings have been confirmed and strengthened and, thus, clearly remain valid.”

(From: WHO, Review of evidence on health aspects of air pollution – REVIHAAP Project, in Technical Report. 2013, WHO European Centre for Environment and Health, Bonn, WHO Regional Office for Europe: Copenhagen, Denmark.)

We can understand that there is a political debate going on in the US and that there is political and economical gain from downplaying the effects of PM2.5, but as scientists we must rely on scientific evidence. If hundreds of studies including both epidemiological and toxicological studies support an association, we cannot adhere to the politics of downplaying the PM2.5 effects based two articles from blogs. The difference between publishing something in a scientific journal and in a blog is that the latter have not been peer-reviewed and, thus, should be read with caution.

The manuscript appears to rely on the environmental standards of nearby countries to argue the relevance of the pollutant concentrations discussed.  But if these standards as well as those in the US are not based on sound science, this is of little relevance. 

We use air quality guidelines supported by sound evidence as reviewed by WHO. These standards are part of our country´s national goal, called the Clean Air objective, created by the Swedish Environmental Protection Agency (EPA) or “Naturvårdsverket” in Swedish.  We hope this is clear throughout the document, if not please tell us where it is unclear so we can understand why you thought it was other countries’ standards.

I do not think this issue needs to result in extensive rewriting of the manuscript, just mention and discussion in the introduction of the paper and perhaps elsewhere as a major uncertainty in the conclusions of the paper.

We have considered the reviewers input and have added a sentence to support the scientific evidence behind PM2.5 effects on Page 2: 

Line 58: "Moreover, the World Health Organization’s comprehensive 2013 Review of evidence on health aspects of air pollution (REVIHAAP) provide confirmation of a causal link between PM2.5 exposure specifically and adverse health outcomes including morbidity and mortality, cardiovascular disease, childhood respiratory disease, cognitive development and, most pertinent to this study, birth outcomes [30]. With this, particulate matter’s short-and long-term effects on human health are reinforced and their continued investigation warranted."

New reference:

30. WHO, Review of evidence on health aspects of air pollution – REVIHAAP Project, in Technical Report. 2013, WHO European Centre for Environment and Health, Bonn, WHO Regional Office for Europe: Copenhagen, Denmark.
